# Learning Visual Parkour from Generated Images

**Alan Yu\*, Ge Yang[1]\*, Ran Choi, Yajvan Ravan, John Leonard, Phillip Isola**
\*Equal contribution.   MIT CSAIL, [1]Institute for AI and Fundamental Interactions

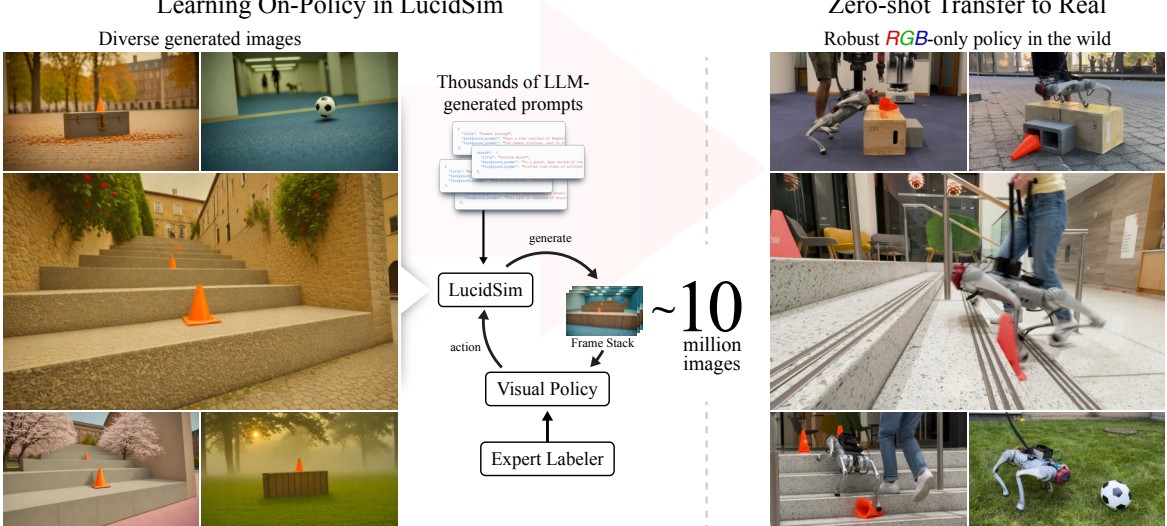

Figure 1: Learning a real-world policy from generated images. *Left*: we source structured image prompts from an LLM, which are combined with a depth map and semantic masks to produce *diverse* and *on-policy* visual data. *Right*: the policy is sufficiently robust to transfer to a variety of challenging terrains in the real world, despite never having seen real data during training.

**Abstract:** Fast and accurate physics simulation is an essential component of robot learning, where robots can explore failure scenarios that are difficult to produce in the real world and learn from unlimited on-policy data. Yet, it remains challenging to incorporate RGB-color perception into the sim-to-real pipeline that matches the real world in its richness and realism. In this work, we train a robot dog in simulation for visual parkour. We propose a way to use generative models to synthesize diverse and physically accurate image sequences of the scene from the robot's ego-centric perspective. We present demonstrations of zero-shot transfer to the RGB-only observations of the real world on a robot equipped with a low-cost, off-the-shelf color camera.   *project website*: https://lucidsim.github.io

## 1   Introduction

The success of a robot learning system depends largely on the realism and coverage of its training data. Real-world data, though inherently realistic, is limited in its coverage over the diverse scenarios a robot might encounter upon deployment. Real training data typically only includes a small number of environments and is not a reliable source for failures that cause injury or harm. As our robot improves throughout training, the data it needs to improve its skills further also evolves. Getting the right data is critical for improving the robot's performance, but in current practice, this is a manual process that needs to be repeated from scratch for new scenes and new tasks.

The alternative is to train in simulations, where we can sample a greater diversity of environmental conditions, and our robots can safely explore failure cases and learn directly from their own actions. Despite substantial investment into simulated physics and rendering, our best efforts at achieving

8th Conference on Robot Learning (CoRL 2024), Munich, Germany.

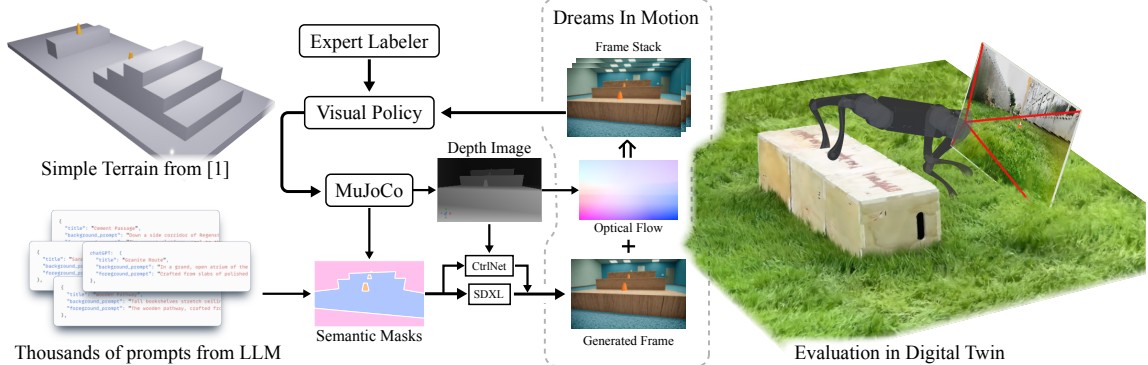

Figure 2: The LucidSim graphics pipeline. We use the same parameterized terrain geometry as [5]. We use MuJoCo to simulate the physics, and render semantic masks and the depth image that are then fed into a ControlNet trained with MiDAS depth maps. The generated image is then combined with the dense optical flow to generate short videos via Dreams In Motion (DIM, see Sec. 2).

realism retain a reality gap [1, 2, 3, 4]. This is because rendering a realistic image means making detailed and realistic scene content. Trying to hand-craft such contents at scale to obtain the diversity required by our robots for sim-to-real transfer is prohibitively expensive. Without diverse and high quality scene content, robots trained in simulation are too brittle to transfer to the real world. Therefore, how to match the real world in its infinite jest, and integrate color-perception into sim-to-real learning, is a key challenge.

The purpose of this work is to develop a solution. We turn to generative models as a promising new data source for robot learning and use visual parkour as a testbed, where a robot dog equipped with a single color camera is tasked to scale tall obstacles at a fast speed. Our ultimate vision is to train robots entirely in generated worlds. At the heart of this is finding ways to exert precise control over the semantic composition and scene appearance to align with the simulated physics—while maintaining the randomness critical for sim-to-real generalization.

Our method works as follows (Figure 2): we take a popular physics engine, MuJoCo [2], and render the depth image and semantic masks at each frame, that together are used as input to a depth-conditioned ControlNet. We then compute the ground-truth dense optical flow from the known scene geometry and changes in the camera poses, and warp the initial generated frame for the following six timesteps to produce a temporally consistent video sequence. On the learning side, we train the visual policy in two stages: first, we optimize the policy to imitate expert behavior from rollouts collected by a privileged teacher. The policy performs poorly after this pre-training step. The post-training step involves collecting on-policy data from the visual policy itself, interleaved with learning on all data aggregated so far. Repeating this step three times makes the visual policy significantly more performant. This policy is sufficiently robust to transfer zero-shot to color observations in the real world throughout our test scenes.

Our contributions are threefold: First, we provide a scalable recipe to translate compute into real-world capabilities by producing geometrically and dynamically aligned visual data for robots. Second, we propose an auto-prompting technique to significantly increase data diversity, which in practice also enables tailored data synthesis. Finally, we provide the first demonstration of a robust, visual parkour policy trained entirely in simulation that has seen zero real-world data.

## 2   LucidSim: Generating Diverse Visual Data with Physics Guidance

We consider a *sim-to-real* setup, where the robot is trained in simulation and transferred to the real world without further tuning. We have partial knowledge about the environments we intend to deploy our robot in—perhaps a rough description or a reference image. Since this information is incomplete, we rely on prior knowledge within generative models to fill the gap. We refer to this

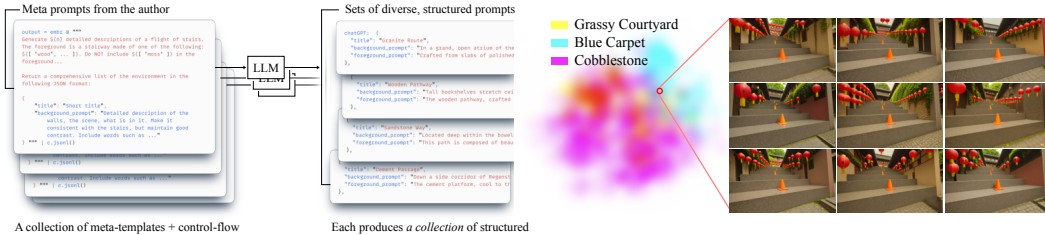

Meta prompts from the author

A collection of meta-templates + control-flow

Sets of diverse, structured prompts

Each produces *a collection* of structured prompts for image generation.

Grassy Courtyard
Blue Carpet
Cobblestone

Figure 3: We solicit batches of $20 \sim 30$ image prompts in JSON format from chatGPT. Each task requires $\sim 10^3$ generated prompts.

Figure 4: CLIP embeddings of images generated by three sets of meta-prompts. Images from the same prompt (○) are not diverse.

guided process as Prior-Assisted Domain Generation (PADG), that begins with an auto-prompting technique critical for synthesizing diverse domains.

**Sourcing diverse, structured prompts from LLM.** We observed early on that repeatedly sampling from the same prompt (Figure 4) tends to reproduce similar-looking images. To obtain diverse images, we first generate batches of structured image prompts by prompting chatGPT with a "meta" prompt that contains a *title block*, *details of the request*, and ends with a question asking for *structured outputs* in JSON (3). Our request includes particular weather, time of day, lighting conditions, and cultural sites. It is impractical to edit the generated image prompts by hand. Instead, we tweak the meta prompts by generating a small number of images, and iterate until they consistently produce reasonable images. Examples of diverse samples from the same meta-prompt, but different image prompts, are shown on the bottom row of Figure 5.

**Generating images with geometry and physics guidance.** We augment a vanilla text-to-image model [6] with additional semantic and geometric control that aligns it with simulated physics. First, we replace the text prompt for the image with pairs of ***prompts*** and ***semantic masks*** that each correspond to a type of asset (Figure 2). In the stairs scene, for instance, we specify the material and texture of the steps inside a coarse silhouette via text. To make the images geometrically consistent, we take an off-the-shelf ControlNet [7] trained on monocular depth estimates from MiDAS [8]. The conditioning depth image is computed by inverting the z-buffer and normalizing it within each image. It is important to adjust the control strength to avoid losing image details (discussed in depth in Section 4.6). Our scene geometry is simple terrain sourced from prior work [5, 9, 10] that optionally includes side walls. We avoid randomizing the terrain geometry to focus our analysis on visual diversity.

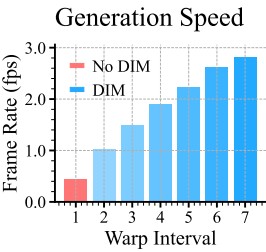

Generation Speed

Figure 6: A $6.5\times$ speed up in video generation.

To produce short videos, we developed **Dreams In Motion (DIM)**, which warps a generated image into subsequent frames using the ground-truth optical flow computed from the scene geometry and the change in

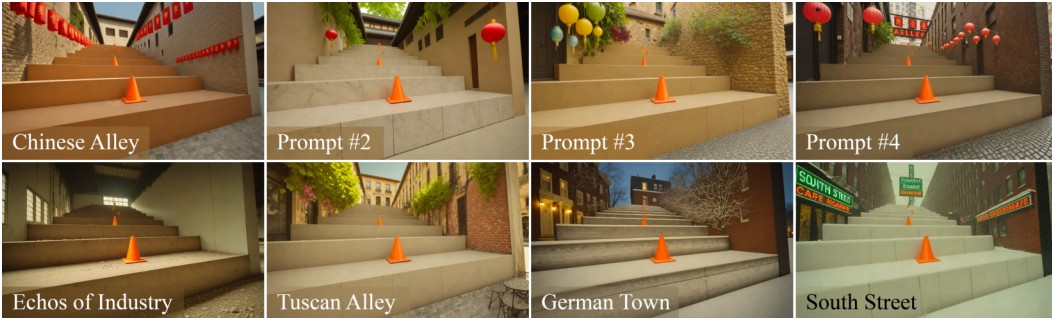

Chinese Alley | Prompt #2 | Prompt #3 | Prompt #4

Echos of Industry | Tuscan Alley | German Town | South Street

Figure 5: LucidSim image samples from the stairs environment. *Top row*: images generated from different prompts produced by the same meta prompt; *bottom row*: different meta prompts.

the camera perspective between two the frames (see Figure 2). The resulting image stack contains timing information critical for parkour. Generation speed is also important. DIM significantly improves the rendering speed (Figure 6) because computing the flow and applying the warping is significantly faster than generating images.

## 3 Learning Robust Real-world Visual Policy from On-Policy Supervision

Our training procedure has two phases: a *pre-training* phase that bootstraps the visual policy by imitating a privileged expert that has direct access to a height map, trained via RL following the procedure in [5]. We collect rollouts from the expert and its imperfect earlier checkpoints, and query the expert for action labels to supervise the visual policy. The visual policy performs poorly after pre-training, but it makes sufficiently reasonable decisions for collecting on-policy data in the second, *post-training* phase (Figure 7). We follow DAgger [11] and combine the on-policy rollouts with the teacher rollouts from the previous step. We collect action labels from the expert teacher and run seventy epochs of gradient descent with the Adam optimizer [12] under a cosine learning rate schedule. We repeat the DAgger iteration three times. This second stage is responsible for the majority of the final performance, approaching expert-level.

**A simple transformer policy.** We present a simple transformer architecture that reduces the number of moving parts for working with multi-modal inputs (Figure 8). Prior work in quadruped parkour uses a composite architecture that first processes the depth image into a compact latent vector using a ConvNet, followed by a recurrent backbone [5]. We use a five-layer transformer backbone with the multi-query attention (MQA [13]). The input camera feed is diced into small **patches** and processed in parallel by a convolution layer. We then stack these tokens together with a linear embedding of the **proprioceptive observation** of the same time step. We repeat this for all timesteps and add a **learnable embedding** at the token level. We find that for RGB images, it is helpful also to include a batch normalization layer before the convolution. We compute the **action output** via an additional class token (cls) stacked at the end of the input sequence, followed by a ReLU latent layer and a linear projection.

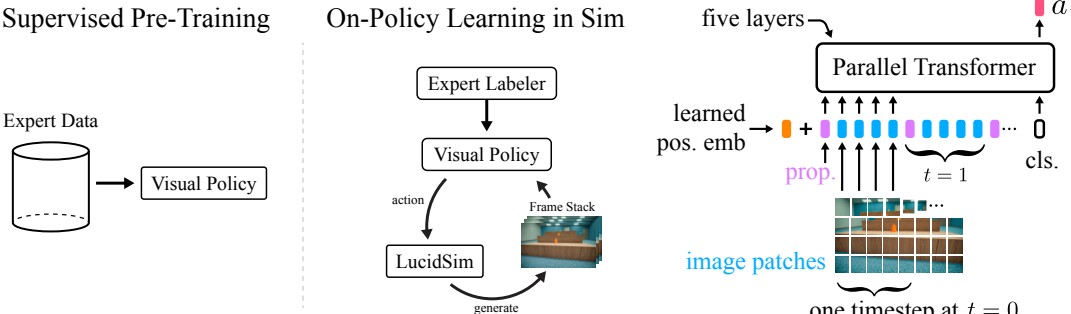

Figure 7: *Pre-training*: Expert rollouts are labeled offline using LucidSim's graphics pipeline. *Post-Training in Sim*: Data is collected on-policy with the visual policy, supervised by action labels from the expert.

Figure 8: We treat **proprioceptive** observations and **image patches** as tokens, plus an additional class (cls) token for the **action output**.

This five-layer transformer policy can process seven input frames while maintaining a 50 Hz frame rate when running on the Nvidia AGX Orin. This memory span is rather short (140 ms). Common ways to speed up LLM inference do not apply due to the use of a rolling input window. This is a key bottleneck that affects skills that require a longer memory. For instance, the robot needs a memory span closer to 400 ms in order to jump over wide gaps.

## 4 Results

We consider the following tasks: tracking a soccer ball (**chase-soccer**); tracking an orange traffic cone (**chase-cone**), climbing over hurdles (**hurdle**); and traversing stairs featuring various material

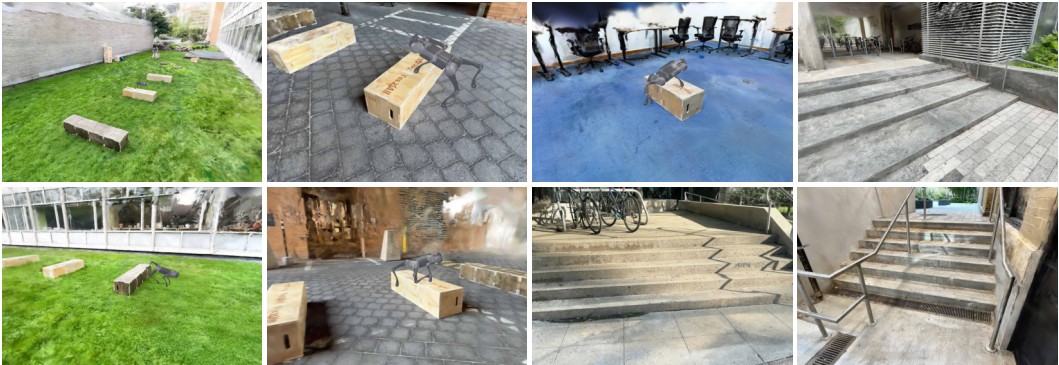

Figure 9: Rendered views of a subset of environments used for evaluation. Each scene is modeled using 3D Gaussian Splatting. The first-person view from the robot's perspective is highly realistic.

types (**stairs**). We evaluated the performance of our learned controller in both the real world and on a small set of real-world scenes modeled in simulation using 3D Gaussian Splatting [14], a recent graphics technique for building complex and photo-realistic environments transferred from the real-world. Examples of these benchmark environments are shown in Figure 9.

In chasing tasks, we randomly sampled locations for the target objects within the robot's camera frustum. For hurdle and stairs, we manually placed orange cones to visually indicate the waypoints. Each task was evaluated in three replica scenes with 50 trials each, randomizing both the starting pose and waypoint location offsets. We report the following metrics in Table 1 and 6: fraction of goals reached (FGR) computed via the ratio $\frac{\text{goals reached}}{\text{goals total}}$, and normalized forward displacement towards the *final* goal, computed via $x_{\text{dist.}} = \frac{x_{\text{reached}}}{\text{distance to goal}}$.

We consider the following baselines: an expert policy that requires privileged terrain data (the oracle); a depth student policy trained using the identical pipeline; an RGB student policy trained using classical domain randomization over textures, and our method, LucidSim, trained with generated frame stacks using DIM. We also provide the simulated performance of the Extreme Parkour [5] depth policy for calibration, which is trained on a magnitude more data. Details on the distinction between the depth baselines can be found in Section 4.4.

## 4.1 Learning from Generated Images Out-Performs Domain Randomization

In our simulated evaluations, we observed that LucidSim outperformed classical domain randomization [15] in almost all evaluations, as seen in Tables 1 and 6. The domain randomization baseline was able to climb stairs quite effectively in simulation, likely due to the repetitive gait that was induced after recognizing the first step. However, it struggled to perform on hurdles, where the timing of the jump was critical. The depth student suffered from subtle but common sim-to-real gaps in the 3D scene. For instance, the oracle policy struggled in one of the stairs environments (Marble) due to the presence of a railing, which it had never seen before in its training environment. We found that the LucidSim policy was less affected by it. Similar phenomena also affected the depth student,

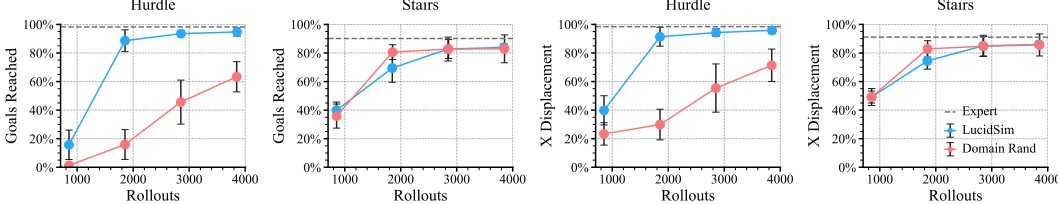

Figure 10: On-policy supervision significantly boosts performance. Each data point represents a new DAgger step. Increasing the number of DAgger iterations improves performance on the simulated benchmark environments. Evaluation includes 50 unrolls on three environment instances for each task. Gray dotted line indicates the performance of the expert teacher.

which was distracted by features such as chairs, walls, and railings in the benchmark environment. Past work used aggressive clipping to mitigate this type of sim-to-real gap, which we elaborate on in Section 4.4.

Table 1: Success Rate (Fraction of Goals Reached) In Real-to-Sim Benchmark Environments.

| Method | Obs. Space | Chase-Cone | | | Chase-Soccer | | | Hurdle | | | Stairs | | |
|---|---|---|---|---|---|---|---|---|---|---|---|---|---|
| | | Lawn | Lab | Urban | Lawn | Lab | Urban | Lawn | Lab | Urban | Bricks | Concrete | Marble |
| Privileged Expert | state+terrain | 98.6 | 96.2 | 97.9 | 98.6 | 96.2 | 97.9 | 95.8 | 100.0 | 99.0 | 97.0 | 100.0 | 73.4 |
| Extreme Parkour [5] | clipped depth | – | – | – | – | – | – | 82.3 | 84.0 | 98.5 | 97.0 | 93.5 | 86.0 |
| Depth | depth | 80.7 | 80.7 | 80.7 | 80.7 | 84.7 | 80.0 | 78.3 | 56.0 | 54.0 | 93.0 | 86.0 | 72.9 |
| Depth | clipped depth | 98.7 | 87.3 | 98.7 | 98.7 | 92.7 | 98.7 | 70.7 | 83.7 | 84.7 | 94.0 | 85.0 | 85.4 |
| Domain Rand. | color | 81.9 | 50.4 | 66.7 | **97.3** | 76.7 | 78.0 | 56.5 | 52.5 | 44.0 | **95.5** | **81.5** | 71.7 |
| LucidSim | color | **96.7** | **84.0** | **98.0** | 88.7 | **90.7** | **94.7** | **90.7** | **93.5** | **96.5** | 87.0 | 81.0 | **83.7** |

## 4.2 Transfer Zero-shot to The Real World

**Experiment setup:** We deployed LucidSim on a Unitree Go1 equipped with a budget RGB webcam and ran inference on the Jetson AGX Orin. Each task was evaluated in multiple scenes, and we recorded whether the robot reached the target object (chase) or successfully traversed the obstacle.

We compare LucidSim to Domain Randomization (DR) and present the results in Figure 11. In the chasing tasks, we observed that Domain Rand. was able to identify color well (orange cones), but struggled with recognizing the patterns of the soccer ball. On the other hand, LucidSim was not only able to recognize the classic black and white soccer ball, but also generalized to different colored soccer balls due to the rich diversity of the generated data it had seen before.

| Task | Trials | LucidSim | Domain Rand. | Depth |
|---|---|---|---|---|
| **cone** | 20 | 100.0% | 70.0% | 80.0% |
| **soccer** | 20 | 85.0% | 35.0% | 65.0% |
| **dark hurdle** | 15 | 86.7% | 26.7% | 86.7% |
| **light hurdles** | 15 | 73.3% | 40.0% | 80.0% |
| **stairs** | 10 | 100.0% | 50.0% | 100.0% |

Figure 11: We measured the success rate of the LucidSim, Domain Rand., and Depth student in a variety of real-world scenarios. Each task was evaluated over multiple environments, diverse in appearance.

For hurdles and stairs, Domain Rand. failed to consistently recognize the obstacle in front of it, often resulting in a head-on collision, while LucidSim was able to consistently anticipate the obstacle and successfully traverse it. We additionally evaluated the clipped depth baseline and show that LucidSim achieves comparable results.

## 4.3 Learning On-Policy Out-Performs Naïvely Scaling Expert Data

We compare learning from on-policy data against naïvely scaling expert data collection in Figure 12. The performance gain from training on additional expert-only data saturated quickly. In both domains, on-policy learning through DAgger was necessary for producing a sufficiently robust policy. The discrepancy was especially apparent on hurdles, where the student struggled to make any meaningful progress through the terrain. We observe the benefit of DAgger with both LucidSim and the domain randomization baseline in Figure 10, with LucidSim reaching a higher overall performance.

## 4.4 Depth-Only Policy Overfits to Training Geometry

Besides Extreme Parkour [5], we consider two depth policies, both trained identically to LucidSim, but with different depth inputs. The first (row three in Tables 1 and 6) receives depth up to a far clip of 5m and shares the same $120°$ FoV as the color methods. The second (row four in Tables 1 and 6, indicated by the clipped depth observation space) receives depth clipped to 2m, as in [5]. We also reduce the FoV and set a near clip of 0.28m to agree with the properties of the Realsense camera. In our simulated evaluation (Tables 1 and 6), we observed that the policy trained with unclipped depth overfits to the minimal and simple geometry of the training scenes, getting thrown off by distractors in the background of the evaluation scenes. The depth policies with limited vision did not suffer as much from the diversity in the test scenes and achieved significantly higher performance.

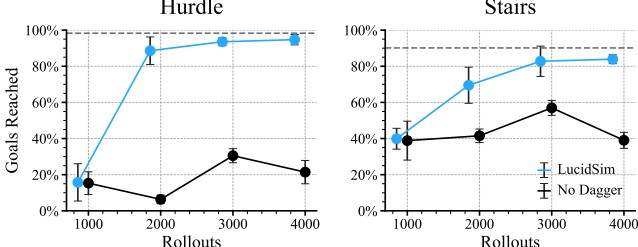

Figure 12: We compare learning on policy against naïve scaling of expert data. Increasing the amount of expert data does not improve performance significantly.

Figure 13: Warped images (blue) increase speed without degrading policy performance.

We interpret this as a failure mode for training on depth—without limiting the vision of the depth policy, the diversity in the test scenes confuses the student. LucidSim is less susceptible to this, as it adds diversity to the policy's experience by hallucinating the background. Although limiting vision is beneficial here for parkour tasks, there is no general way to know which part of the visual input to censor for which task, and coming up with effective censoring strategies currently requires hand design. In particular, if the task required tracking more distant objects, then the 2m clip would be inappropriate.

### 4.5 Understanding the Speed and Performance of Dreams in Motion (DIM)

Image generation is a bottleneck in our pipeline. DIM greatly accelerates each policy unroll, while also providing dynamically-consistent frame stacks by trading off diversity. We study how the student's performance is affected by generating every frame independently, as opposed to warping batches of images with DIM. We consider the Hurdle domain as it is the most challenging. As illustrated by Figure 13, the performance was similar, yet DIM was able to achieve the same results in a fraction of the time (Figure 6).

### 4.6 Strong Conditioning Decrease Diversity and Image Details

There exists a trade-off between the accuracy of the geometry versus the richness of the details in the generated images. When the conditioning strength is too low, the image deviates from the scene geometry (left, Figure 14). When it is too strong, the image loses the diversity and rich details (right side of Figure 14) and becomes highly reduced due to over-constraints.

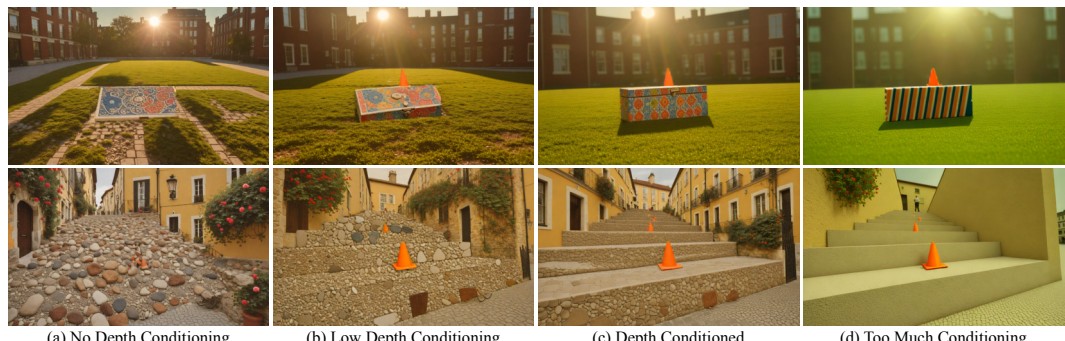

(a) No Depth Conditioning     (b) Low Depth Conditioning     (c) Depth Conditioned     (d) Too Much Conditioning

Figure 14: Stronger geometric control reduces the level of detail. As we increase the control strength from left to right, the geometry becomes better aligned with the depth map. However, this comes at the cost of reducing the amount of details in the image. Prompts in Appendix A.7.

## 5 Related Work

**Generative AI for task-creation and world-building.** LucidSim is part of a growing line of work in robot learning that uses generative AI systems to automatically design parts of the learning setup,

including the task specification [16, 17], and the reward function [18]. This is distinct from work that augments real-world data with image in-painting [19, 20], or style transfer [21]. Works that learn action-conditioned video generation models [22, 23] are also being considered as a potential source of unlimited data [24, 25, 26]. In general, however, it is challenging for video models to learn accurate physics [27]. LucidSim's hybrid approach aligns with a growing trend in frontier video generation models that use physics-based image manipulation for guidance, followed by generative in-painting [28].

**Robot parkour.** Recent work in agile locomotion uses deep reinforcement learning and supervised distillation in simulated environments to achieve impressive levels of agility in quadrupeds [5, 29, 9] and humanoid robots [30]. These methods all rely on depth images as input. In contrast, our work does not depend on depth and uses the color image from a low-cost, off-the-shelf webcam instead. We choose visual parkour because a blind teacher can not accomplish this task [31]. To our knowledge, LucidSim is the first reported result of visual parkour using a color camera and the first that is trained entirely in simulation with generated images.

**Robot learning from demonstrations.** Recent work in robot learning leverages low-cost hardware and expressive new policy classes borrowed from language modeling and image generation to produce increasingly capable visuomotor controllers [32, 33, 34]. More recent work further lowers the barrier-to-scale by removing the robot, retaining just the end effector [35, 36, 37]. The remaining cost is dominated by the need to visit and set up physical scenes and the human effort in tailoring data collection to the evolving robot. In contrast, LucidSim aims to move data collection from the physical world into software.

**Real-to-sim and learning from digital twins.** New techniques [38, 14] in computer graphics have made it easy to build high-fidelity digital twins in sim. Efforts in drone-racing [39], autonomous-driving [40], and humanoid soccer [41] take advantage of this to produce robust but highly specialized controllers. In contrast, we employ these techniques for evaluation only, where targeted assessment via a few high-quality digital scans can be highly effective.

# 6 Conclusion

We present a technique to synthesize unlimited, geometrically, and dynamically correct, multi-frame image stacks for robot learning. We also provide the first empirical demonstration of a visual parkour policy on a quadruped robot trained entirely using generated data. Although preliminary, we consider these results a promising proof-of-concept that points towards a more common-place usage of generative AI in producing learning data for difficult robotic tasks.

**Key Assumptions and Limitations.** Enforcing the geometry through conditioning degrades the richness of the generated images (Section 4.6). A way to apply loose control [42] would be helpful. We also made terrain randomization outside the scope of this work and opted to use simple hand-designed geometries inherited from prior work [43, 44, 10]. Ultimately, we believe the 3D assets and the scene layout will also be generated autonomously [43, 44], where AI models amplify human creativity and effort.

# Acknowledgments

This work was partially supported by a Packard Fellowship and a Sloan Research Fellowship to P.I., by ONR MURI grant N00014-22-1-2740, by Amazon.com Services LLC Award #2D-06310236, and by the Defence Science and Technology Agency, Singapore. This work was also partially supported by ONR grant N00014-19-1-2571 (Neuroautonomy MURI) and by the MIT Lincoln Laboratory. G.Y. was partially supported by the National Science Foundation Institute for Artificial Intelligence and Fundamental Interactions (IAIFI) under the Cooperative Agreement PHY-2019786.

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

# Appendix

## A.1 Generative Workflow

We developed our image generation workflows with ComfyUI [45], a popular software tool with a graphical user interface that facilitates rapid prototyping. Figure A15 contains a screenshot of the stairs scene. The supplementary material includes a copy of the Python implementation of this workflow that we use in production to generate images for this paper.

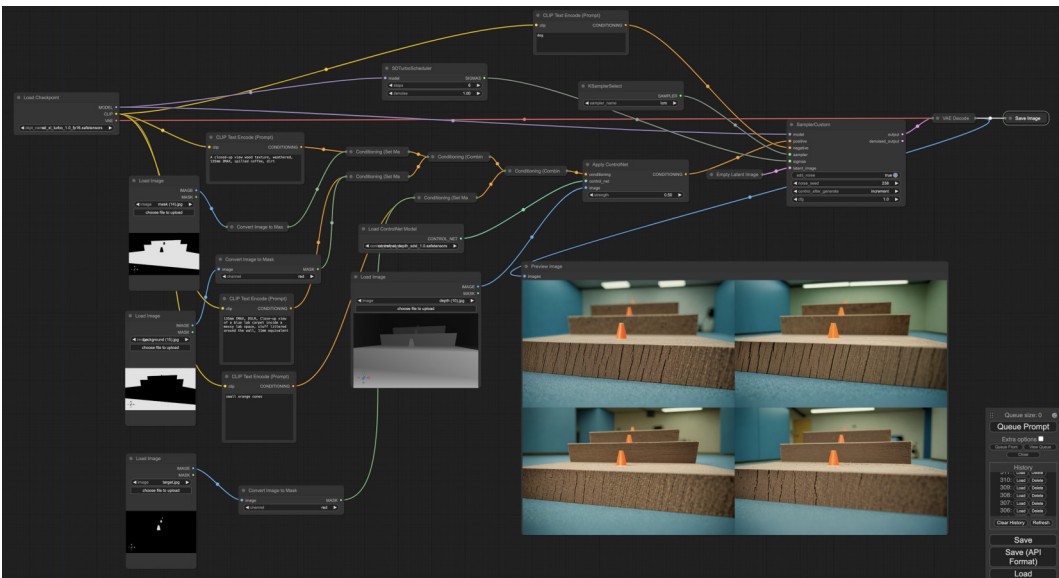

Figure A15: Generative workflow. This example is for the stair scene.

A key feature of this workflow is Area Composition, where we use semantic masks to limit the cross-attention with a text prompt to specific regions. The base generative model, Stable Diffusion XL Turbo [6], when used in combination with the LCM sampler [46], is very fast and can generate reasonable images with a single diffusion step. Image quality improves with more diffusion steps, up to eight or ten, beyond which it starts to degrade. We run six diffusion steps in production to balance between speed and image quality.

## A.2 System Design Strategies for Scaling Image Generation

Image generation contributes the bulk of each experiment's wall-clock time. We accelerate data generation by distributing trajectory sampling and image generation to parallel workers. We set up two Zaku task queues [47]. The first one is responsible for dispatching unroll requests for each trajectory to the trajectory sampling workers. The image generation node is called "dream-weaver," or *weaver* in short. The second Zaku task queue is responsible for dispatching image rendering requests to these weaver workers. We present an overview of the system design in Figure A16.

The system design is different between the pre-training phase using expert unrolls, and the on-policy learning step that samples from the visual policy. In the pre-training phase (see Figure A16 a), we render images after each rollout. The unroll workers upload the semantic masks, optical flow, and depth image to the weaver queue, which triggers the weaver workers to generate and upload the requested image to a centralized data server.

In the on-policy learning phase, we need to sample actions from the visual policy, which requires image observations as input. We implemented pub-sub and remote procedural calls (RPC) in Zaku (see documentation), backed by a redis replica set. At the start of each flow stack (of seven frames), the unroll worker dispatches a request for the first frame to the weaver RPC queue. Once the generated

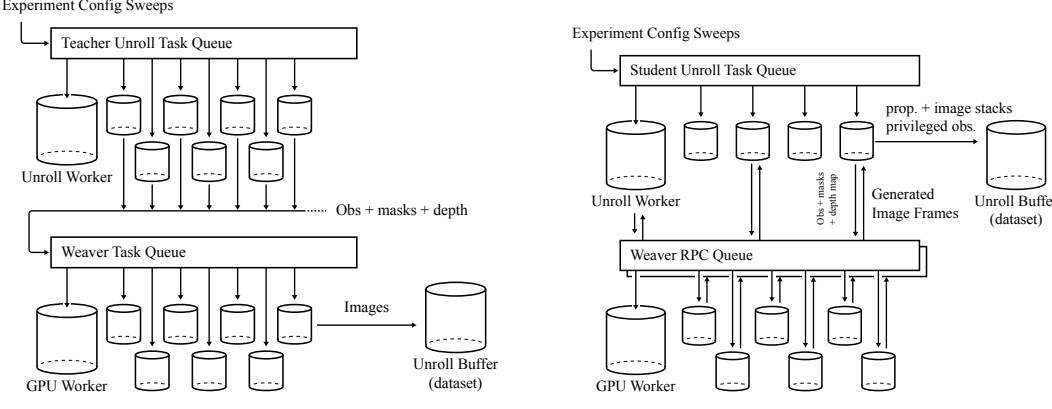

(a) Generating Visual Data Off-line.          (b) Collecting On-Policy Unrolls Requires RPC.

Figure A16: System Architecture. *Left*: collecting unroll data from the expert teacher does not require generating the images online. We populate the *weaver queue* once after collecting a batch of teacher unroll, and the weaver workers upload the generated and downsized images to the ml-logger service. *Right*: collecting unroll from the visual policy required building remote-procedural call (RPC) into the task queue. The unroll buffer is a centralized ml-logger server that stores and serves the unroll data for the learning step.

frame has been returned it is used as input to the visual policy. During the subsequent six timesteps, the unroll worker computes the optical flow and uses to warp the initial frame into those following frames. This process restarts after six steps, with a new weaver RPC request.

## A.3  Domain Randomization Baseline

Our domain randomization pipeline [15] is adapted from Robosuite [48]. We randomize the appearance of the terrain by sampling textures (solid, checker, noisy, gradient), color, and material

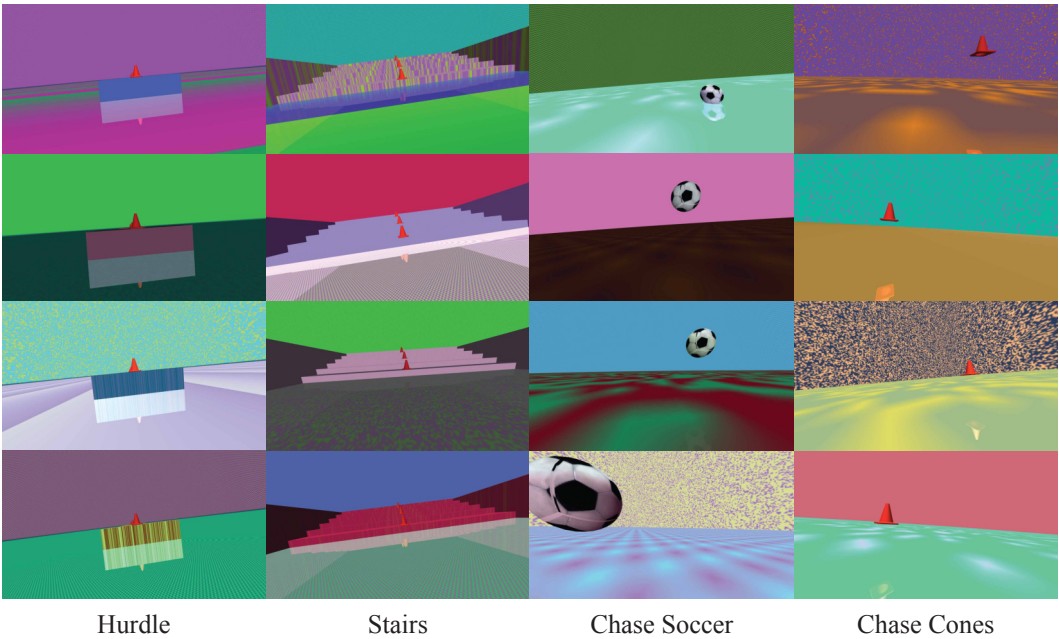

Hurdle          Stairs          Chase Soccer          Chase Cones

Figure A17: Sample images from the Domain Randomization baseline. Textures, colors, material properties, and lights are randomized every seven steps. We do not randomize the cone so that the policy can learn to use it as a landmark. This makes it a fair comparison to LucidSim.

properties (reflectance, shininess, specular). We also randomize the lighting parameters of each light in the scene. Just as with LucidSim, we sample a new appearance every seven frames. Figure A17 includes samples of the rendered image from the domain randomization baseline on all four domains.

## A.4 Training Details

The expert teacher is trained according to the procedure described in [5]. We reproduce the reward used during training in Table 2; the PPO training parameter in Table 3; and the domain randomization in Table 4.

For imitation learning (both phases), we use the same training hyper-parameters presented in Table 5. In the pre-training phase, we sampled the teacher trajectories from a mixture of expert teachers of multiple random seeds and their intermediate checkpoints. The action labels used to supervise the visual policy were always computed by the expert teacher.

Table 2: Expert Reward Terms

| Term | Symbol | Scale |
|------|--------|-------|
| parkour velocity tracking [5] | $\min(\langle \boldsymbol{v}, \hat{\boldsymbol{d}}_w \rangle, v_{cmd})$ | 1.5 |
| yaw tracking | $\exp\{-|\omega_z - \omega_z^{\text{cmd}}|\}$ | 0.5 |
| z velocity | $v_z^2$ | -1.0 |
| roll-pitch velocity | $|\omega_{xy}|^2$ | -0.05 |
| base orientation | $\mathbb{1}_{\text{flat}}|\boldsymbol{g}_{xy}^{\text{proj}}|^2$ | -1.0 |
| hip position | $|\boldsymbol{q}_{\text{hip}} - \boldsymbol{q}_{\text{hip}}^0|^2$ | -0.5 |
| collision | $\mathbb{1}_{\text{collision}}$ | -10.0 |
| action rate | $|\boldsymbol{a}_t - \boldsymbol{a}_{t-1}|^2$ | -0.1 |
| joint accelerations | $|\ddot{\boldsymbol{q}}|^2$ | -2.5e-7 |
| delta joint torques | $|\boldsymbol{\tau}_t - \boldsymbol{\tau}_{t-1}|^2$ | -1.0e-7 |
| joint torques | $|\boldsymbol{\tau}|^2$ | -1e-05 |
| joint error | $|\boldsymbol{q} - \boldsymbol{q}^0|^2$ | -0.04 |
| foot vertical contact [5] | $\sum_i \mathbb{1}_{\text{vertical contact}}^{(i)}$ | -1.0 |
| foot clearance [5] | $\sum_i \mathbb{1}_{\text{edge contact}}^{(i)}$ | -1.0 |

Table 3: Expert Training Parameters

| Hyperparameter | Value |
|----------------|-------|
| value loss coefficient | 1.0 |
| clip range | 0.2 |
| entropy coef | 0.01 |
| learning rate | 2e-4 |
| # minibatches per epoch | 4 |
| # epochs per rollout | 5 |
| # timesteps per rollout | 24 |
| discount factor | 0.99 |
| GAE parameter | 0.95 |
| max grad norm | 1.0 |
| optimizer | Adam |
| joint stiffness | 20 |
| joint damping | 0.5 |

Table 4: Expert Randomization Parameters

| Term | Min | Max | Unit |
|------|-----|-----|------|
| friction range | 0.6 | 2.0 | - |
| added mass | 0.0 | 3.0 | kg |
| Body Center of Mass | -0.20 | 0.20 | m |
| push velocity $(v_x, v_y)$ | 0.0 | 0.5 | m/s |
| Motor Strength | 80 | 120 | % |
| Forward Velocity Command $(v_x)$ | 0.3 | 0.8 | m/s |

Table 5: Behavior Cloning Parameters

| Hyperparameter | Value |
|----------------|-------|
| max. timesteps per rollout | 600 |
| rollouts per DAgger Iteration | 1000 |
| learning rate | 5e-4 |
| timesteps | 70 |
| optimizer | Adam |
| weight decay | 5e-4 |
| momentum | 0.9 |
| dropout | 0.1 |

## A.5 Real-to-Sim Evaluation Environments

Figure A18 provides an overview of our process for constructing the evaluation environments. For each task, we select a few scenes that differ in appearance (e.g. red bricks, pavement, grass, indoors). We report the results from three different scenes on each task in Tables 1 and 6. We capture $\approx 500$ images for each scene, and extract the resulting collision mesh from Polycam. For appearance, we run COLMAP [49, 50] to obtain camera pose estimates, and reconstruct the scene using 3D Gaussian Splatting (3DGS) [14, 51, 52].

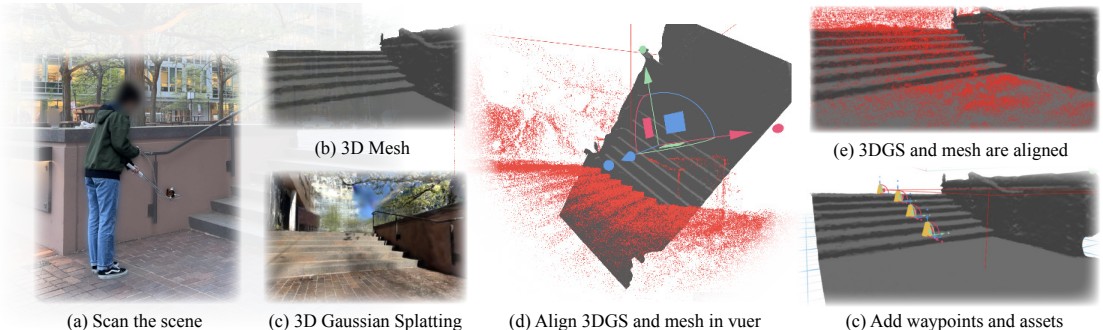

(a) Scan the scene      (b) 3D Mesh      (c) 3D Gaussian Splatting      (d) Align 3DGS and mesh in vuer      (e) 3DGS and mesh are aligned      (c) Add waypoints and assets

Figure A18: Process for making the benchmark environments. (a-c) Scan the scene to collect the 3D mesh and the 3D Gaussian Splats. (d-e) These two are initially unaligned, so we manually scale and align them. (f) add the markers for the waypoints.

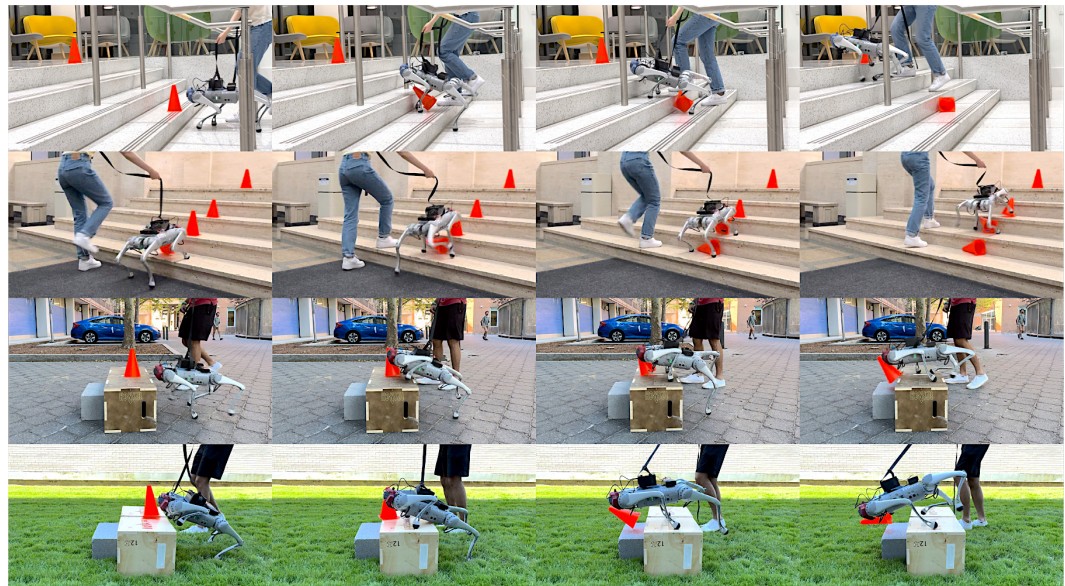

Figure A19: Learning Visual Parkour from Generated Images. Top to bottom: (1,2) robot climbing stairs. (3) robot climbing hurdles on a stone ground (4) on a grassy courtyard. Notice the different box color. For more experiment videos, visit https://lucidsim.github.io

We use our custom viewer to align the collision mesh with the gaussian splat. For the hurdle and stairs scenes, we manually label 3-5 waypoints along the course that appear as orange traffic cones. We use the collision mesh as the terrain, and the 3DGS render as visual observation to the robot. For objects that are not present in the initial scan (i.e., soccer ball, traffic cones), we apply the mask rendered by the physics engine to insert them into the robot's ego view.

## A.6 Prompt Used in Figure 4

**Foreground prompt** : "Cool, gray slabs of granite that are flecked with darker mineral deposits. The polished finish is unmarred but faintly glistening under the ambient light, revealing a durable, ancient presence."

**Background prompt** : "An ancient alley lined with tea houses and small, quaint shops, each displaying traditional ornaments and calligraphy. The walls are adorned with ivy and red paper decorations, while overhead, strings of lanterns sway gently in the breeze."

## A.7  Prompts Used in Figure 14

Hurdle scene "Sunny Afternoon"

**foreground** : "A ceramic box with colorful, intricate patterns."

**background** : "The sun illuminates a somewhat unkempt lawn dotted with dry patches. Gravel paths crisscross the grass, leading to sunlit, red-brick buildings with large, gleaming windows."

Stairs scene "Cemented Courtyard Pathway"

**foreground** : "Composed of a myriad of small, smooth pebbles incorporated within, creating a uniquely textured appearance. Discoloration and cracks reflect its rich history of countless treaders."

**background** ": "Walls of natural stone, varying from reds to yellows, provide a strikingly authentic atmosphere. Wrought iron railings draped with climbing roses frame the scene. The melody of a nearby street musician overlays the hum of quiet conversations, while the occasional cyclist clacks down the cobbled thoroughfare."

## A.8  X-Displacement

We report the corresponding X-Displacement metric for Table 1 in this section for completeness.

Table 6: **X-Displacement In Real-to-Sim Benchmark Environments.**

| Method | Obs. Space | Chase-Cone | | | Chase-Soccer | | | Hurdle | | | Stairs | | |
|---|---|---|---|---|---|---|---|---|---|---|---|---|---|
| | | Lawn | Lab | Urban | Lawn | Lab | Urban | Lawn | Lab | Urban | Bricks | Concrete | Marble |
| Privileged Expert | state+terrain | 99.6 | 99.1 | 98.7 | 99.6 | 99.1 | 98.7 | 96.3 | 100.0 | 99.0 | 97.2 | 100.0 | 76.0 |
| Extreme Parkour [5] | clipped depth | – | – | – | – | – | – | 85.9 | 88.8 | 98.8 | 96.6 | 95.5 | 83.2 |
| Depth | depth | 95.8 | 93.6 | 93.8 | 95.0 | 92.9 | 92.9 | 80.7 | 70.4 | 59.1 | 93.5 | 88.8 | 76.5 |
| Depth | clipped depth | 99.9 | 94.7 | 99.8 | 99.8 | 97.9 | 99.9 | 75.4 | 88.5 | 89.5 | 94.0 | 88.7 | 86.1 |
| Domain Rand. | color | 91.6 | 81.2 | 84.9 | **99.3** | 89.2 | 89.5 | 66.6 | 61.6 | 57.1 | **95.4** | 85.1 | 76.5 |
| LucidSim | color | **99.5** | **92.7** | **99.7** | 92.3 | **96.8** | **98.0** | **92.6** | **94.0** | **97.8** | 88.8 | **85.6** | **83.6** |

