# OpenReview forum: "Learning Visual Parkour from Generated Images"
_robot-learning.org/CoRL/2024/Conference — CoRL 2024_

### Official Review · Reviewer_n6VN · 2024-07-18
**Promising work for scaling locomotion through generative AI.**

**Originality:** 4
**Technical Quality:** 4
**Clarity Of Presentation:** 2
**Potential Impact:** 3
**Recommendation:** 3
**Confidence:** 3

**Review:**

# Strengths
- Experiments are performed to show improved behavior compared to a depth baseline, which has been the de-facto input for related works (L185-186).
- Promise of scaling locomotion with generative AI.
- Lots of optimizations are performed to improve the speed of training (DIM and DAgger w/ multiple iterations).
- Experiments are performed over multiple tasks.

# Weaknesses
- Some technical details are lacking.
   - In Section 2, a problem formulation is introduced. It would be beneficial if the method in Section 3 directly referenced and utilized this formulation, including specific training details.
   - Giving a technical formula or explanation for FGR and X-Displacement would be very helpful.
   - L82: Is yaw-pitch-roll referring to the joints or the base frame?

- Readability could be improved.
   - Including references to the figures within the text would significantly increase their usefulness.
   - Page numbers appear to be missing, adding them would improve navigation through the document.

**Quality Of The Limitations Section:**

3

**Questions For Rebuttal:**

1) Add the depth model as a baseline in the real-world experiments table (Figure 11).
2) Add the requested technical details to the methods and experiments section.

(Post-Rebuttal). I thank the authors for their improvements and additional experiments to the manuscript. It appears that the table pasted below may have some inaccuracies, as the table for reviewer di9g seems to have more accurate numbers based on the reply. It might also be beneficial if the methods section variables were utilized more explicitly, rather than just introduced and condensed.

**Robotics Focus:**

4

**Summary Of Paper:**

This work attempts to learn visual parkour by training with a teacher-student method and performing behavior cloning using generative AI. To do so, the work proposes using a meta-prompt to utilize ChatGPT to generate prompts to create diverse images for training. Dreams-in-motion is then proposed to generate short videos for training, used in conjunction with DAgger. Results are demonstrated on multiple tasks, where notable baselines include a standard augmentation to RGB as well as a commonly utilized depth-based model. Results are shown in both real-world and real-to-sim experiments.

**Summary Of Recommendation:**

This work demonstrates a shift from recent work that has performed parkour abilities through depth, instead a RGB-based method is proposed to do this. The proposed method outperforms a domain randomization and depth-based model as baselines. With the combination of adding scaling through generative AI, this method is promising for improving related locomotion works.

---

### Official Review · Reviewer_di9g · 2024-07-21
**Novel approach to bridging RGB sim-to-real gap, but limited to known environments**

**Originality:** 3
**Technical Quality:** 4
**Clarity Of Presentation:** 3
**Potential Impact:** 3
**Recommendation:** 3
**Confidence:** 3

**Review:**

Quality: The work is generally high-quality. The authors have done a good job solving the numerous problems that are prerequisites of the proposed approach (e.g. how do you generate multiple diverse RGB images corresponding to a single depth trajectory? etc.) and have done a lot of work training in simulation and evaluating in the real world, and from their videos it is clear that the proposed method works well. They have clearly spent a lot of time on the diagrams etc. too and the paper is overall a pretty good presentation. It would be nice to have some experiments that compare the RGB policy against a policy that is trained and tested using depth observations to show that the approach provides comparable results on cheaper hardware.

Clarity: I have some concerns about the clarity of the writing - it took me two full reads from start to finish to understand the exact problem/solution (problem: RGB images from simulation are not good for sim-to-real transfer. solution: feed simulation's depth image to a diffusion model to generate diverse RGB images that all show a physically similar situation, train on those instead). It does sound in places like the authors are generating the whole scene from scratch in simulation and not just coloring it because the discussion of the pre-designed scene geometry is mostly avoided. How exactly the experiments are performed (e.g. what simulator, where did the scene geometry come from, etc.) is not described at all and thus the reader has to fill some of these gaps themselves.

Originality: As far as I'm aware, using the depth images to imagine alternative RGB observations using a diffusion model is a novel approach to bridging the sim-to-real gap. This is combined with some now-established methods from CV like meta-prompting to generate diffusion model prompts.

Significance: The model does help bridge the part of the sim-to-real gap caused by rendering/material/texture etc. issues and probably provides some robustness against domain shifts. However, it still relies on the scene geometry being known ahead of time (e.g. the simulator needs to have the same scene modeled), which is costly and straight out impossible for novel scenes encountered at test time. This greatly blunts the potential usefulness of the approach: it is only relevant for when you want to train an RGB policy (rather than a depth one) on a scene you already have a rough digital twin of.

**Quality Of The Limitations Section:**

3

**Questions For Rebuttal:**

Can you describe the exact simulation setup (e.g. what simulator, how did you design the geometry, etc.)?

Can you also describe some possible approaches for how the need for hand-designing the scene geometry be avoided?

How do you think the policy performs compared to a policy that is trained and tested with depth images instead of RGB?

**Robotics Focus:**

4

**Summary Of Paper:**

The paper introduces a set of approaches that can be used to generate realistic RGB observations from simulated depth images to use when training a RGB-input policy, helping avoid the low real-world accuracy of policies trained directly on simulated RGB images.

**Summary Of Recommendation:**

The paper introduces a clever new approach to bridging the sim-to-real gap caused by poor rendering. However, it has some strong assumptions (the scene geometry needs to be modeled ahead of time) which lowers the significance of the contribution. Nonetheless, it is an interesting contribution that would be nice to see at CoRL.

---

### Official Review · Reviewer_xbt2 · 2024-07-24
**LucidSim: Learning Agile Visual Locomotion from Generated Images**

**Originality:** 5
**Technical Quality:** 2
**Clarity Of Presentation:** 2
**Potential Impact:** 4
**Recommendation:** 3
**Confidence:** 3

**Review:**

This paper demonstrates how to use generative AI to generate a diversity of experience to support RL-based training of a walking robot. It works with chatGPT to generate a generator of environment description, which is then used to generate a physically consistent sequence of rendering.

The overall concept is very interesting and presented with a very clear global description of the pipeline. However, the authors chose to provide a large number of unusually large figures, which uses a lot of space in the paper. This provides a qualitative evaluation of the generation. The space used by these figures would be put to a better use by including a more thorough description of the pipeline and design choice (a lot of it is presented in the appendices). Extensive illustrations could be moved to the appendices instead.

The quantitative evaluation is made through the evaluation of the use of the generated sequences in a RL algorithm.

The paper is interesting as a tutorial but lacks a level of insight I would expect from a scientific contribution: for instance the text of the ablation study is shorter than the space used by the "ablation study" heading. This is not an ablation study.

**Quality Of The Limitations Section:**

1

**Questions For Rebuttal:**

The ablation study and the limitation section are both only a few lines and most of the pipeline is described in the appendices. Because CoRL is a scientific conference, I feel that the large space allocated to figures should be moved to the appendices, and the paper itself should focus on the scientific contribution, its algorithms, its detailed analysis and justification, with a real ablation study.

**Robotics Focus:**

4

**Summary Of Paper:**

This paper demonstrates how to use generative AI to generate a diversity of experience to support RL-based training of a walking robot. It works with chatGPT to generate a generator of environment description which is then used to generate physically consistent sequence of rendering.

**Summary Of Recommendation:**

This paper presents a very interesting new idea that could prove to be a major contribution to the RL community. Nevertheless, it is not presented with the thoroughness expected from a scientific contribution.

---

### Author Rebuttal · Authors · 2024-08-14

Please see the attached file for the revised manuscript.

---

### Decision · Program_Chairs · 2024-09-04

**Decision:**

Accept

**Comment:**

High level summary of reviews (PRE REBUTTAL):

Strengths:

- Reviewer di9g highlights that the paper is a novel and interesting idea. Other reviewers rate originality quite high (if not explicitly calling out originality in the review); there are no obvious issues around lack of technical contribution.
- The proposed method is shown to be effective in generating diverse and realistic training data for RL-based learning of walking robots. It is clear that the proposed method works well.
- The paper is well-written, with generally good figures and reasonably clear explanations of the proposed method and experimental results (with some caveats around completeness and clarity - see below).

Weaknesses:

- Some reviewers found that the paper lacks sufficient technical details and could benefit from a more thorough description of the pipeline and design choices. Improving the clarity of writing (see reviewer suggestions for specifics) will help with this.
- The level of details for the ablation study is considered insufficient and could be improved to provide more insights into the contribution of different components of the proposed method.
- One reviewer suggests that the paper could be improved by comparing the proposed method with relevant literature.
- The significance of the contribution is questioned by one reviewer, who argues that the approach has strong assumptions (e.g., the need for pre-modeled scene geometry) that limit its usefulness.

POST REBUTTAL:

The authors reworked the introduction, clarified a number of key items requested by the reviewers and added additional experiments. While reviewer ratings didn't change, the paper is still quite a strong submission and is improved following the edits during rebuttal.